# Finding lost DG: Explaining domain generalization via model complexity

## Abstract

The domain generalization (DG) problem setting challenges a model trained on multiple known data distributions to generalise well on unseen data distributions. Due to its practical importance, a large number of methods have been proposed to address this challenge. However most of the work in general purpose DG is heuristically motivated, as the DG problem is hard to model formally; and recent evaluations have cast doubt on existing methods' practical efficacy – in particular compared to a well chosen empirical risk minimisation baseline. We present a novel learning-theoretic generalisation bound for DG that bounds novel domain generalisation performance in terms of the model's Rademacher complexity. Based on this, we conjecture that existing methods' efficacy or lack thereof is a variant of the standard empirical risk-predictor complexity trade-off, and demonstrate that their performance variability can be explained in these terms. Algorithmically, this analysis suggests that domain generalisation should be achieved by simply performing regularised ERM with a leave-one-domain-out cross-validation objective. Empirical results on the DomainBed benchmark corroborate this.

## 1 Introduction

Machine learning systems have shown exceptional performance on numerous tasks in computer vision and beyond. However performance drops rapidly when the standard assumption of i.i.d. training and testing data is violated. This domain-shift phenomenon occurs widely in many applications of machine learning (Csurka, 2017; Zhou et al., 2021; Koh et al., 2021), and often leads to disappointing results in practical machine learning deployments, since data 'in the wild' is almost inevitably different from reference training sets.

Given the practical significance of this issue, a large number of methods have been proposed that aim to improve models' robustness to deployment under a different distribution than used for training (Zhou et al., 2021), a problem setting known as domain generalisation (DG). These methods span diverse approaches such as specialised neural architectures, data augmentation strategies, and regularisers. Nevertheless, the DG problem setting is difficult to model formally for principled derivation and theoretical analysis of algorithms, the target domain of interest is unobservable during training, and cannot be directly approximated by the training domains due to unknown distribution shift. Therefore the majority of these existing approaches are based on poorly understood empirical heuristics.

To make matters worse, a recent study by Gulrajani & Lopez-Paz (2021) assessed the state of DG research with a carefully conducted comparative evaluation of algorithms on a large benchmark suite under a common platform. They found that published methods were not as effective as claimed, and in particular reported that 'no existing method reliably beats a well tuned empirical risk minimization (ERM) baseline'. We argue that this negative result highlights the need for better theory in this area in order to understand why existing algorithms have such erratic performance, and to guide the development of principled algorithms that are more effective and reliable.

To this end, our first contribution is to present an intuitive learning-theoretic bound for DG performance. Intuitively, while the held-out domain of interest is indeed unobservable during training, we can bound its performance using learning theoretic tools similar to the standard ones used to bound the performance on (unobserved) testing data given (observed) training data. In particular we show that the performance on a held out target domain is bounded by the performance on known source

domains, plus two additional model complexity terms, that describe how much a model can possibly have overfitted to the training domains. This theoretical contribution leads to several insights.

Firstly, our theory suggests that DG performance is governed by a trade-off between empirical risk and model complexity that is analogous to the corresponding and widely understood trade-off that explains generalisation in standard i.i.d. learning as an overfitting-underfitting trade-off (Geman et al., 1992). Based on this, we hypothesise that performance variability is determined by implicit or explicit regularisation. That is, the plethora of different strategies available (Zhou et al., 2021) – from data-augmentation to specialised optimisers – actually affect DG performance by explicitly or implicitly choosing different fit-complexity trade-offs. We corroborate this hypothesis by evaluating a number of models in the DomainBed suite in terms of complexity, and showing that their apparently erratic performance in Gulrajani & Lopez-Paz (2021) is actually consistent with an explanation in terms of implied complexity.

Practically, our analysis suggests that the model selection strategy (Hastie et al., 2009) is a factor in DG performance that is at least as important as the actual mechanism of model complexity control (i.e., Tuning of regularisation strength vs specific parametric design of regulariser). In particular, regularisation should be stronger when optimizing for future DG performance than when optimizing for performance on seen domains. Unfortunately, model complexity is hard to carefully control in deep learning due to the large number of relevant factors (architecture, regularisers, implicit regularisation from optimiser, etc). Gulrajani & Lopez-Paz (2021) attempted to address this by hyper-parameter search in the DomainBed benchmark, but are hampered by the computational infeasibility of accurate hyper-parameter search. In this paper, we use linear models and off-the-shelf self-supervised features to demonstrate much more clearly how cross-domain performance depends on complexity. Specifically, our theoretical and empirical results show that, contrary to the conclusion of Gulrajani & Lopez-Paz (2021), simple domain-wise cross-validation is a better objective to drive DG model selection.

In summary, based on our new generalisation bound, and associated empirical analysis, our take-home messages are: (i) Model fit vs complexity trade-off is a key determinant of DG performance, that explains existing DG algorithm performance variability. (ii) The complexity control strategy used to determine bias-variance trade-off is crucial in practice, with peak DG performance achieved when optimizing model complexity based on domain-wise validation. (iii) Regularisation required for optimal DG is greater than for conventional optimization for within-domain performance.

## 2 RELATED WORK

**Theoretical Analysis of the DG Setting and Algorithms**    The DG problem setting was first analysed in Blanchard et al. (2011). Since then there have been some attempts to analyse DG algorithms from a generalisation bound perspective (Muandet et al., 2013; Blanchard et al., 2021; Hu et al., 2020; Albuquerque et al., 2020; Rosenfeld et al., 2021). However these studies have theoretical results that are either restricted to specific model classes, such as kernel machines, or make strong assumptions about how the domains seen during training will resemble those seen at test time—e.g., that all domains are convex combinations of a finite pre-determined set of prototypical domains. In contrast, our Rademacher complexity approach can be applied to a broad range of model classes (including neural networks), and makes comparatively milder assumptions about the relationship between domains—i.e., they are i.i.d. samples from another arbitrary distribution over domains.

The majority of the existing work investigating the theoretical foundations of DG follow the initial formalisation of the domain generalisation problem put forth by Blanchard et al. (2011), where the goal is to minimise the expected error over unseen domains. However, several recent works have also explored the idea of bounding the error on a single unseen domain with the most pathological distribution shift (Janzing, 2019). This type of analysis is typically rooted in methods from causal inference, rather than statistical learning theory. As a consequence, they are able to make stronger claims for the problems they address, but the scope of their analysis is necessarily limited to the scenarios where their assumptions about the underlying causal structures are valid. For example, Janzing (2019) provides bounds that assume problems conform to a specific class of structural equation models, and the analysis is performed under the assumption that infinite training data is available within each of the observed training domains. Throughout the work we address the stan-

dard DG formalisation given by Blanchard et al. (2011), where one is concerned with the expected performance of a model on domains sampled from some distribution over domains.

Others rely on trying to link between domain *adaptation* objectives (where target domains are observable for alignment to source domains) and domain generalisation (where target domains are not observable and thus cannot correctly be used in a learning objective). Albuquerque et al. (2020) proceed by making assumptions on the structure of the distribution over possible domains (i.e., that it has support determined by the convex hull of a finite set of prototypical domains), which allows them to upper bound the domain alignment metric. Ye et al. (2021) provide a bound that depends on an unobservable domain distance quantity, which they then approximate in experiments using kernel density estimates.

Rosenfeld et al. (2021) is another piece of work that theoretically investigates the generalisation of ERM in a DG setting. They deal with online DG, where each time-step corresponds to observing a new domain, and the learner must produce a new model capable of generalising to novel domains. Another point of difference between their work and the standard DG problem setting of Blanchard et al. (2011) is that the domain at each time-step is chosen by an adversary. They analyse this game for a finite number of time-steps, but they assume each domain has an infinite amount of data. They also put some limitations on the adversary: e.g., it must choose a domain that is a convex combination of a finite number of pre-determined domains. In contrast, our theoretical analysis is in the more realistic setting where one has a finite amount of data per domain, and the domains we consider are not limited to convex combinations of a set of prototypical domains. Possibly the most similar work to our theoretical contributions is due to Ahuja et al. (2021), who also provide learning-theoretic generalisation bounds for DG. However, their analysis only applies to finite hypothesis classes (which does not include, e.g., linear models or neural networks), whereas ours can be applied to any class amenable to analysis with Rademacher complexity.

**Empirical Analysis** The main existing empirical analysis on DG is Gulrajani & Lopez-Paz (2021), who compared several state of the art DG methods under a common evaluation and hyper-parameter tuning protocol called DomainBed. They ultimately defend Empirical Risk Minimization (ERM) over more sophisticated alternatives on the grounds that no competitor consistently beats it across the benchmark suite. We also broadly defend ERM, and build on the same benchmark, but differently we provide a much deeper analysis into when and why ERM works. More specifically: (i) We provide a new theoretical analysis of ERM's generalisation quality unlike the prior purely empirical evaluation, (ii) We re-use the DomainBed benchmark to directly corroborate this theory under controlled conditions using linear models where model complexity can be tractably and accurately tuned. (iii) We use our complexity-based analysis to explain the previously erratic results of prior DomainBed competitors in terms of model complexity. (iv) We identify, and empirically validate, the preferred model selection criterion for DG, a point which was inconclusive in Gulrajani & Lopez-Paz (2021).

## 3 BOUNDING RISK FOR DOMAIN GENERALIZATION

**I.i.d. learning** is concerned with learning a mapping from some input space $\mathcal{X}$, to a label space $\mathcal{Y}$, given data drawn from a distribution on $\mathcal{X} \times \mathcal{Y}$. One aims to find a model, $f^* \in \mathcal{F}$, that minimises the expected loss (also called risk) on unseen data,

$$f^* = \arg\min_{f \in \mathcal{F}} L_p(f), \qquad L_p(f) = \mathbb{E}_{(\vec{x},y) \sim p}[\ell(f(\vec{x}), y)], \tag{1}$$

where $p$ is the data distribution and $\ell(\cdot, \cdot)$ is the loss function. In practice, we only have access to a finite set of data, $S = \{(\vec{x}_i, y_i)\}_{i=1}^m$, sampled i.i.d. from this distribution, so must minimise an empirical risk estimate,

$$\hat{f} = \arg\min_{f \in \mathcal{F}} \hat{L}_p(f), \qquad \hat{L}_p(f) = \frac{1}{m} \sum_{i=1}^m \ell(f(\vec{x}_i), y_i). \tag{2}$$

where $m$ is the number of training examples. One of the central focuses of statistical learning theory is to bound the difference between these two types of risk. For example, in the standard single domain setting this can be done via

$$L_p(f) \le \hat{L}_p(f) + 2\mathcal{R}_m(\mathcal{F}) + \mathcal{O}\left(\sqrt{\frac{\ln(1/\delta)}{m}}\right), \tag{3}$$

which holds with probability at least $1 - \delta$, and $\mathcal{R}(\mathcal{F})$ is known as the empirical Rademacher complexity of the hypothesis class, $\mathcal{F}$. This complexity term is defined as

$$\mathcal{R}_m(\mathcal{F}) = \mathbb{E}_{\vec{\sigma}} \left[ \sup_{f \in \mathcal{F}} \frac{1}{m} \sum_{i=1}^{m} \sigma_i \ell(f(\vec{x}_i), y_i) \right]. \tag{4}$$

For a hypothesis class consisting of norm-constrained linear classifiers, one can achieve the following upper bound on the empirical Rademacher complexity (Shalev-Shwartz & Ben-David, 2014),

$$\mathcal{R}_m(\mathcal{F}) \leq \frac{XB}{\sqrt{m}}, \qquad \mathcal{F} = \{\vec{x} \mapsto \vec{x} \cdot \vec{w} : \|\vec{w}\|_2 \leq B\},$$

where we assume that each input, $\vec{x}$, has a Euclidean norm of at most $X$. There are a variety of ways to define hypothesis classes for neural networks, but most recent approaches take the view of fixing a particular architecture and specifying constraints on the norms of weights or distances they can move from their initialisations (Bartlett et al., 2017; Neyshabur et al., 2019; Gouk et al., 2021).

**Domain generalisation.** While standard i.i.d. learning assumes all data come from the same distribution, the DG problem setting assumes the existence of an *environment* $\mathcal{E}$, of distributions $p$ (Blanchard et al., 2011). Note that we do not restrict what types of differences one could see between different domains sampled from $\mathcal{E}$: for any two distributions in $p, q\text{supp}(\mathcal{E})$ it could be the case that either $p(\vec{x}) \neq q(\vec{x})$, or $p(y|\vec{x}) \neq q(y|\vec{x})$, or even that both types of distribution shift have occurred. The conceptually simplest—and often implicitly assumed—goal of DG methods is to minimise the expected risk across different distributions that could be sampled from the environment,

$$L^{\mathcal{E}}(f) = \mathbb{E}_{p \sim \mathcal{E}}[L_p(f)]. \tag{5}$$

This object is also the most commonly analysed idealised objective in the learning theory literature (Blanchard et al., 2011; Muandet et al., 2013; Blanchard et al., 2021; Rosenfeld et al., 2021), but other formulations also exist (Arjovsky et al., 2019). As with the single domain learning problem, we only have access to an empirical estimate of the risk,

$$\hat{L}^{\mathcal{E}}(f) = \frac{1}{n} \sum_{j=1}^{n} \hat{L}_{p_j}(f), \tag{6}$$

where we assume for ease of exposition that all $n$ domains have the same number of examples.

### 3.1 BOUNDING THE GENERALISATION GAP

We next bound the generalisation gap between the observed empirical risk, $\hat{L}^{\mathcal{E}}$, on the source domains and the expected risk, $L^{\mathcal{E}}$, on unseen domains that holds uniformly for all hypotheses in $\mathcal{F}$.

**Theorem 1.** *For a 1-Lipschitz loss, $\ell(\cdot, \cdot)$, taking values in $[0, 1]$, with confidence at least $1 - 2\delta$ for all $f \in \mathcal{F}$ we have that*

$$L^{\mathcal{E}}(f) \leq \hat{L}^{\mathcal{E}}(f) + 2\mathcal{R}_{mn}(\mathcal{F}) + 2\mathcal{R}_n(\mathcal{F}) + 3\sqrt{\frac{ln(2/\delta)}{2mn}} + 3\sqrt{\frac{ln(2/\delta)}{2n}},$$

*where $n$ is the number of training domains and $m$ is the number of training examples in each training domain.*

The proof can be found in Appendix A.

**Discussion** Theorem 1 tells us that expected risk on unseen domains is bounded by the empirical risk (training loss) on seen domains, plus Rademacher complexity terms and sampling error terms that decay with the number of domains and training instances per domain. As with typical single-domain bounds, the latter sampling error terms are not under control of the model designer. The former Rademacher terms describe the complexity of the chosen model class and govern how much it could possibly overfit to the seen domains while minimising empirical risk. In the case of linear models, these terms depend on weight norms, while in the case of deep models they further depend on properties of the chosen network architecture. Mirroring conventional generalisation in standard i.i.d. learning, a very simple model may minimise the Rademacher terms, $\mathcal{R}$, while producing high

empirical risk, $\hat{L}^{\mathcal{E}}$, and vice-versa. Thus good generalisation critically depends on a carefully chosen empirical risk vs model complexity trade-off. The difference between Theorem 1 and single domain bounds (e.g., Equation 3) is the additional dependence on the number of domains $n$ and additional Rademacher complexity term. This implies that when the goal is to generalise to new domains, the risk of overfitting is higher. Therefore we expect a lower complexity model to me optimal for held out domain performance compared to seen domain performance in standard i.i.d. learning.

## 3.2 BOUNDING THE EXCESS RISK OF ERM

As a second theoretical contribution, we next bound the excess expected risk between the ERM solution $\hat{f}$ and the *best possible model* $f^*$ within the function class $\mathcal{F}$. Note that bounding excess risk, as opposed to the generalisation gap, overcomes some of the known issues with theoretically analysing the generalisation properties of deep networks (Belkin et al., 2018).

**Corollary 1.** *With probability as least $1 - 2\delta$, the excess risk of the empirical risk minimiser in $\mathcal{F}$ is bounded as*

$$\mathbb{E}[L^{\mathcal{E}}(\hat{f}) - L^{\mathcal{E}}(f^*)] \leq 2\mathcal{R}_{mn}(\mathcal{F}) + 2\mathcal{R}_n(\mathcal{F}) + 2\sqrt{\frac{ln(2/\delta)}{2mn}} + 2\sqrt{\frac{ln(2/\delta)}{2n}},$$

The proof (in Appendix B) uses the same technique as the usual proof for excess risk (see, e.g., Pontil & Maurer (2013)), and some of the intermediate results required for Theorem 1.

**Discussion** Corollary 1 tells us that the gap between ERM and the *best possible* predictor in the function class depends on the same complexity terms observed in Theorem 1. In particular, for any typical hypothesis class, ERM converges to the optimal classifier at a rate of $\mathcal{O}(1/\sqrt{mn}) + \mathcal{O}(1/\sqrt{n})$. To justify itself theoretically, any future DG method that claims to be better than ERM should either: (1) demonstrate a faster convergence rate than this—at least by an improved constant factor; or (2) formally show that the chosen hypothesis class is composed of models that can extrapolate to new domains without additional data. The latter would likely involve making specific assumptions about the underlying data generation process, coupled with analysis of a specific hypothesis class using methods from causal inference. An example of such analysis is given by Janzing (2019). Methods based on causal inference have the potential to give bounds with much better convergence rates than the rate given above. However, because one must make assumptions about the underlying family of structural equation models, the applicability of such bounds is much more restricted than our Rademacher complexity technique, which does not require these assumptions.

## 4 EXPERIMENTS

Based on our previous theoretical analysis, we conduct experiments on DomainBed (Gulrajani & Lopez-Paz, 2021). In particular we aim to address the following questions: (1) Theorem 1 shows that novel domain generalisation performance is governed by empirical-risk complexity trade-off. *Can we directly demonstrate the dependence of generalisation on complexity by controlling for model complexity?* (2) *Can the erratic performance of state of the art methods previously observed on DomainBed be explained in terms of model complexity?* (3) Given that model complexity is a crucial determinant of generalisation, *what is the best objective for tuning regularisation strength?*

## 4.1 DOMAIN GENERALISATION PERFORMANCE DEPENDS ON MODEL COMPLEXITY

To directly investigate the impact of model complexity on domain generalisation performance, we first work with Linear SVM applied to pre-computed deep features[1]. In this function class model complexity can be directly controlled by a scalar hyper-parameter, the objective is convex so confounding factors in deep network training (optimiser choice, early stopping, etc) disappear, and training is fast enough that we can densely and exhaustively evaluate a wide range of complexities.

---

[1]Note that when a *fixed* feature extractor, or any other preprocessing is used, it does not impact the model complexity or associated generalisation bound.

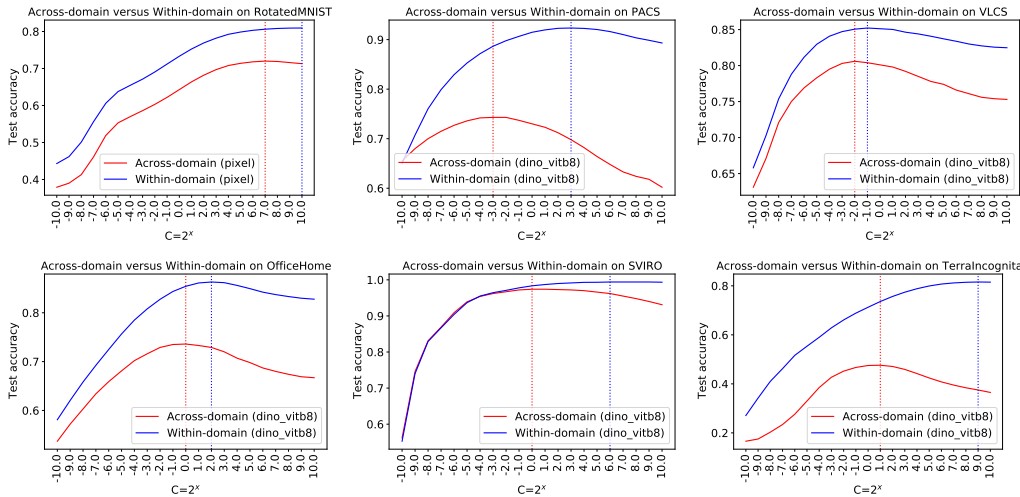

Figure 1: Linear SVC performance on DomainBed benchmark datasets is governed by model complexity parameter $C$. Optimal tuning for performance on novel target domains (DG condition, red) always requires stronger regularisation (lower $C$) than for performance on seen domains (blue).

**Setup**  We use DINO (Caron et al., 2021) pre-trained models, such as DINO-ViTB8 and DINO-ViTS8 to extract features, and then train LinearSVC classifiers. We experiment on six different DG benchmarks, including RotatedMNIST (Ghifary et al., 2015), PACS (Li et al., 2017), VLCS (Fang et al., 2013), OfficeHome (Venkateswara et al., 2017), SVIRO (Dias Da Cruz et al., 2021) and Terra Incognita (Beery et al., 2018). Different trials have different numbers of training instances in practice, so we used a linear SVM objective of the form $\frac{C}{n}\sum_i^n \ell(f_w(x_i), y_i) + \|w\|^2$, with parameter $C$ controlling the complexity of model $w$ through loss-regularisation trade-off. We searched a wide range of $\log_2 C$ in $\{-10, \ldots, 10\}$. We then conduct two experiments, holding training and test split size constant across both: (i) Training on the train splits of all domains, and testing on the test splits of all domains (i.e., standard i.i.d. learning). (ii) Training on the train splits of three domains, and testing on the test splits of a held out domain (i.e., DG performance).

**Results**  The results in Fig 1 average over 5 random seeds for dataset splitting, and all choices of hold-out target domain. From these we can see that: (i) all experiments exhibit the classic trade-off between fitting the data well and constraining hypothesis class complexity appropriately. We observe underfitting for high regularisation (small C), and overfitting at low regularisation (large C). (ii) Comparing the within- and across-domain evaluation: across-domain leads to lower performance—which is to be expected, due to the distribution shift. A more noteworthy observation is that, the optimal regularisation for novel-domain performance is stronger than for seen-domain performance (red vertical lines left of blue). This corroborates our theoretical result that the ideal model complexity is lower for DG than for conventional i.i.d. learning. Note that these results are using an exhaustive oracle to explore mode complexity, and do not constitute a complete algorithm, which would need to chose a specific complexity. We discuss an algorithm in Sec 4.3.

## 4.2 EXPLAINING DOMAINBED DG PERFORMANCE IN TERMS OF COMPLEXITY

Our experiment in Section 4.1 showed that linear models exhibit a very clean trade-off between complexity and generalization. However there are many subtle factors that influence effective model complexity in neural networks besides architecture: learning rate, learning rate schedule, weight decay, early stopping, etc; and the repeatably of learning is hampered by non-convexity, stochastic initialization, and stochastic mini-batches, etc. The previous evaluation study of Gulrajani & Lopez-Paz (2021) attempted some degree of control over these factors for state of the art neural DG methods by applying a common hyperparameter search strategy that aimed to find the best tuning (cf: Fig 1) for each competitor. They nevertheless found existing methods to perform unreliably. Our conjecture is that the number of seed samples (3) and hyper-parameter search trials (20) used was far too few to reliably tune the deep networks evaluated, and therefore the tuning of each model

was inaccurate and performance estimates unreliable. Since dense and accurate hyper-parameter tuning for neural networks is more challenging than our linear models in Section 4.1, we take a different approach. Rather than trying to control complexity directly, we take trained DomainBed methods and *measure their complexity retrospectively*.

**Setup**    To ensure that complexity can be measured accurately, we work with 2-layer MLPs. Specifically, we take fixed ImageNet pre-trained ResNet-18 features, and feed them to MLPs, which are then trained using the DomainBed framework. 2-layer MLPs are sufficient to instantiate many of the state of the art DG algorithms in DomainBed. We train and compare ERM, CORAL (Sun & Saenko, 2016), Mixup (Wang et al., 2020), MMD (Li et al., 2018), RSC (Huang et al., 2020), SD (Pezeshki et al., 2020), VRex (Krueger et al., 2020), and IRM (Arjovsky et al., 2019) with two hyperparameter choices. We also report the results of an ERM model, checkpointed at a range of training iterations.

**Measuring Model Complexity**    Because we have limited our attention to 2-layer MLPs, we can take advantage of a model capacity metric that is specialised for this class of models. We retrospectively determine the complexity of a trained network using the measure proposed by Neyshabur et al. (2019), who also use this measure to bound the Rademacher complexity of a 2-layer MLP hypothesis classes. More concretely, the expression used for computing complexity is

$$\mathcal{C}(f_{U,V}) = \|V\|_F(\|U - U^0\|_F + \|U^0\|_2), \tag{7}$$

where $U$ is the weight matrix of the first layer, $U^0$ is its random initialisation, and $V$ is the weight matrix of the second layer. We use $\|\cdot\|_F$ to denote Frobenius norm, and $\|\cdot\|_2$ to indicate the spectral norm. Note that for simplicity we have omitted constant factors that depend only on the architecture and problem setting, and not the learned weights, as we use the same architecture for all methods we investigate.

**Results**    The results in Figure 2 summarise the trade-off between measured model complexity (x-axis), and held-out domain test accuracy (y-axis), averaged over all choices of held out domain for each dataset (See Supplementary for full breakdown across all held-out domains). The top plot compares several of the published neural DG methods implemented in DomainBed. The main message of the top plot is that results are consistent with the hypothesis that the resulting complexity of the models trained by various DG methods is a key factor in determining domain generalisation accuracy (compare Fig. 2 with the clean result for linear where models we are able to intervene and control complexity directly in Fig. 1)—thus explaining the previously erratic behaviour observed in domain-bed.

To provide a different view of the same issue, the bottom plot reports a vanilla untuned ERM model check-pointed every 300 training iterations between (up to a total of 15,000) but expressed in the *same* complexity units as above. Because neural models can gain complexity with iterations (see, e.g., Prechelt (1998); Hardt et al. (2016)) we also see typical overfitting-underfitting trade-off curves. This shows that, as expected, proper choice of early stopping criterion is important. It also shows that over a similar dynamic range of complexity (0.5-3), ERM (below) and alternative models (above) span a similar dynamic range of accuracy (e.g., 6% for PACS). This suggests that complexity, as controlled by whatever mechanism, is a key determinant of DG performance.

**How Do DG Models Control Complexity?**    By introducing different modifications to standard ERM, DG models explicitly or implicitly modify the bias and variance of the function class to be learned. Gulrajani & Lopez-Paz (2021) highlight neural models as being dependent on learning rate, batch size, dropout rate, and weight decay; with other factors being choice of optimiser and weight initialiser, etc. For example, higher dropout rate and weight decay tend to reduce complexity, while some other factors influence complexity in a less transparent way. RSC introduce more sophisticated dropout-like mechanisms, which would be expected to reduce complexity. Meanwhile alignment-based methods like CORAL and MMD effectively add auxiliary losses, which will implicitly affect complexity, but are hard to explicitly link to it. Consistency based methods like IRM and VRex penalise loss variance, which also tends to reduce the generalisation gap in the single task case (Maurer & Pontil, 2009). The specific setting of all the corresponding hyper-parameters (e.g., regulariser strength, dropout rates) influence final model complexity, which we argue is the key determinant of performance.

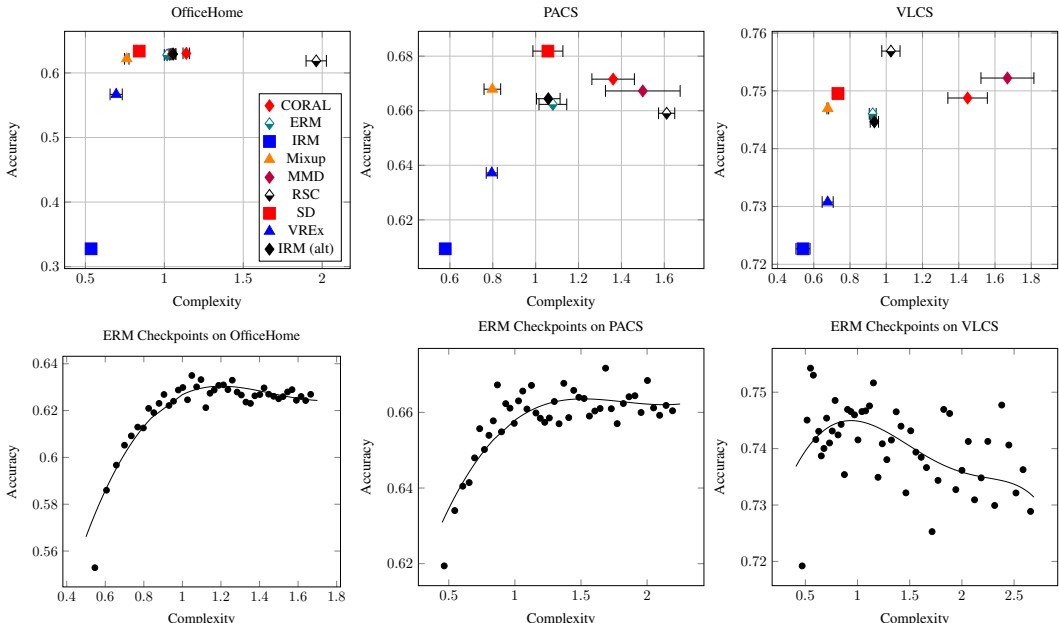

Figure 2: Performance of neural networks as a function of their measured model complexity after training using DomainBed (DB). The overall results are consistent with a standard bias-variance trade-off (cf. Fig 1 for linear models): performance depends on how well each model was tuned by DB's hyper-parameter search procedure. Top: Leave-one-domain-out cross-validation performance of various neural DG algorithms evaluated by DomainBed. Horizontal error bars correspond to the standard deviation of the model complexity measured for each cross-validation iteration. Bottom: Performance of ERM model checkpointed at different training iterations. The central tendency is obtained via fitting a support vector regression model (Shevade et al., 2000) with a 6th order polynomial kernel.

## 4.3 EVALUATING LINEAR MODELS ON DOMAINBED

Having demonstrated control of generalisation with linear model regularisation (Sec. 4.1), and explained DomainBed performance post-hoc via measuring complexity of neural models (Sec. 4.2), we next evaluate algorithms for practical complexity tuning. From Fig. 1 we saw that within-domain and cross-domain evaluation have different optimal tuning. If we consider automating the search for a good regularisation parameter, these two final evaluation criteria correspond respectively to hyper-parameter selection based on a validation set from the seen source domains, vs based on a validation set drawn from a held out source domain. The latter validation criterion corresponds to a unbiased estimator of expected cross-domain generalisation performance $R^{\mathcal{E}}(f)$, which our theorem bounds.

**Setup** We performed DG evaluation on DomainBed using ERM with linear models using the same DINO features as Section 4.1. For each held out domain, we performed hyperparameter tuning with either instance-wise cross-validation (where validation folds are drawn from the same source domains used for training) or domain-wise cross-validation (where validation folds are drawn from held-out domains not seen during training).

**Results** The results in Table 1 report the average accuracy across held out domains, and the average selected $\log C$ regularisation strength (See Supplementary for the full breakdown of results for each held-out domain). We can see that the domain-wise cross-validation objective leads to similar or better accuracy, and similar or smaller C value selection (i.e., stronger regularisation). This outcome is opposite to the observation made by Gulrajani & Lopez-Paz (2021). Given the theoretical support for our domain-wise objective, and our clear empirical outcome when freed of confounding factors such as stochasticity in neural network training, we consider our result to be decisive and attribute the different observation Gulrajani & Lopez-Paz (2021) to the inaccuracy and stochasticity of hyperparameter tuning in neural networks.

| | | Domain Wise | | Instance Wise | |
|---|---|---|---|---|---|
| | | Acc | $\log C$ | Acc | $\log C$ |
| Raw pixel | RotatedMNIST | 71.6 | 4.97 ($\pm$0.52) | 71.2 | 6.82 ($\pm$0.28) |
| DINO VIT-B8 | PACS | 74.3 | -1.73 ($\pm$0.40) | 70.6 | 1.56 ($\pm$0.66) |
| | VLCS | 80.4 | -1.04 ($\pm$0.69) | 80.5 | -0.87 ($\pm$0.35) |
| | OfficeHome | 73.6 | -0.17 ($\pm$0.35) | 73.1 | 1.04 ($\pm$0.40) |
| | SVIRO | 97.2 | 0.21 ($\pm$0.33) | 95.3 | 4.85 ($\pm$0.46) |
| | Terra Incognita | 45.4 | -0.17 ($\pm$0.87) | 38.8 | 5.37 ($\pm$0.35) |
| DINO VIT-S8 | PACS | 73.0 | -1.91 ($\pm$1.43) | 71.6 | 1.04 ($\pm$0.89) |
| | VLCS | 80.4 | -1.91 ($\pm$0.66) | 80.6 | -1.21 ($\pm$0.35) |
| | OfficeHome | 72.2 | -0.69 ($\pm$0.57) | 72.3 | 0.35 ($\pm$0.40) |
| | SVIRO | 94.5 | -1.18 ($\pm$0.47) | 91.3 | 5.68 ($\pm$0.97) |
| | Terra Incognita | 37.1 | 0.00 ($\pm$1.50) | 33.1 | 4.85 ($\pm$0.57) |

Table 1: Comparing model selection criteria using LinearSVC. Accuracy and selected SVM regularisation parameter '$C$'. Domain-wise validation outperforms instance-wise and selects stronger regularisation (lower $C$).

### 4.4 DISCUSSION

At the start of this section we set out to address three questions, which can be answered as: (1) The dependence of cross-domain generalisation on complexity can be directly and precisely when using linear models. (2) The erratic performance of state of the art methods in DomainBed can be largely explained in terms of implied model complexity. (3) If the goal is to optimise for domain generalisation, then then domain-wise validation is preferred to instance-wise validation as a model-selection objective.

Our analysis suggests that several existing methods that tried to tune learnable DG hyper-parameters by performance on a held-out domain (Balaji et al., 2018; Li et al., 2019) were broadly on the right track, and concurs with Gulrajani & Lopez-Paz (2021) that those methods with underspecified hyperparameter and model selection procedures are unhelpful. However, given that most neural methods have many more complexity hyperparameters than the single hyperparameter that we were able to carefully control for linear models, obtaining accurate tuning and reliable performance evaluation is likely to be a challenge. Gradient-based hyper-parameter estimation methods, as initially attempted in Balaji et al. (2018); Li et al. (2019), together with efficient methods for long-inner loop hypergradient calculation (Lorraine et al., 2020), may benefit the former problem by making search in larger numbers of hyperparameters more feasible. Alternatively, using general purpose pre-trained features (Caron et al., 2021) as we did here, and focusing on learning shallow models that can be accurately tuned for DG may be another promising avenue in practice. Although achieving state of the art performance is not our focus, we note that our results in Table 1 are quite competitive with end-to-end trained state of the art (Gulrajani & Lopez-Paz, 2021), despite using fixed features and shallow models.

## 5 CONCLUSION

In this paper we explained the performance of domain generalisation methods in terms of model complexity (bias-variance trade-off) from both theoretical and empirical perspectives. Both perspectives show that complexity impacts cross-domain generalisation in a way that closely mirrors the bias-variance trade-off in conventional i.i.d. learning – but where stronger regularisation is required if optimising for cross-domain generalisation than if optimising for conventional within-domain generalisation. We clarified the preferred model selection criterion in each case. This analysis demystifies the problem being posed by the DG problem setting, why existing algorithms succeed or fail to work, and sets the bar for future theoretical theoretical studies to surpass in terms of convergence rates, as well as for future empirical studies to surpass in terms of strong baselines.

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

## A  PROOF OF THEOREM 1

We use a slightly modified version of the standard empirical Rademacher complexity bound on generalisation error, as stated by Mohri et al. (2018) but originally shown by Bartlett & Mendelson (2002), where the modification is to weaken the i.i.d. assumption to be only an independence assumption—i.e., allow instances to be drawn from different distributions. The proof is exactly the same (because McDiarmid's inequality requires only independence, and not identical distributions), but we re-state it here for completeness.

**Theorem 2.** *For $p_1, ..., p_n$ independent samples from $\mathcal{E}$, and 1-Lipschitz loss $\ell(\cdot, \cdot)$ taking values in $[0, 1]$, the following holds with confidence at least $1 - \delta$,*

$$\frac{1}{n}\sum_{j=1}^{n} L_{p_j}(f) \leq \frac{1}{n}\sum_{j=1}^{n} \hat{L}_{p_j}(f) + 2\mathcal{R}_{mn}(\mathcal{F}) + 3\sqrt{\frac{ln(2/\delta)}{2mn}}, \tag{8}$$

*where $\hat{L}_{p_j}(f)$ is measured on $S_{p_j} = \{(\vec{x}_{ij}, y_{ij})\}_{i=1}^{m}$, a collection of $m$ i.i.d. samples from $p_j$.*

*Proof.* Let $\mathcal{S} = S_{p_1} \bigcup ... \bigcup S_{p_n}$ and define

$$\Phi(\mathcal{S}) = \sup_{f \in \mathcal{F}} \frac{1}{n}\sum_{j=1}^{n}(L_{p_j}(f) - \hat{L}_{p_j}(f)). \tag{9}$$

Note that $\Phi(\mathcal{S})$ satisfies the bounded differences property required by McDiarmid's inequality: i.e., if we construct $\mathcal{S}'$ by replacing any one of the $(x_{ij}, y_{ij})$ in $\mathcal{S}$ with another random variable also drawn from $p_j$, then $|\Phi(\mathcal{S}) - \Phi(\mathcal{S}')| \leq \frac{1}{mn}$. Therefore McDiarmid's inequality implies that with confidence at least $1 - \frac{\delta}{2}$

$$\Phi(\mathcal{S}) \leq \mathbb{E}_{S_{p_{1:n}} \sim p_{1:n}}[\Phi(\mathcal{S})] + \sqrt{\frac{ln(2/\delta)}{2mn}}. \tag{10}$$

We continue by bounding the expected value of $\Phi(\mathcal{S})$,

$$\mathbb{E}_{S_{p_{1:n}} \sim p_{1:n}}[\Phi(\mathcal{S})] \tag{11}$$

$$= \mathbb{E}_{S_{p_{1:n}} \sim p_{1:n}}\left[\sup_{f \in \mathcal{F}} \frac{1}{n}\sum_{j=1}^{n}(L_{p_j}(f) - \hat{L}_{p_j}(f))\right] \tag{12}$$

$$= \mathbb{E}_{S_{p_{1:n}} \sim p_{1:n}}\left[\sup_{f \in \mathcal{F}} \frac{1}{n}\sum_{j=1}^{n}\left(\mathbb{E}_{S'_{p_j} \sim p_j}\left[\frac{1}{m}\sum_{i=1}^{m}\ell(f(\vec{x}'_{ij}), y'_{ij})\right] - \frac{1}{m}\sum_{i=1}^{m}\ell(f(\vec{x}_{ij}), y_{ij})\right)\right] \tag{13}$$

$$\leq \mathbb{E}_{S_{p_{1:n}} \sim p_{1:n}}\mathbb{E}_{S'_{p_{1:n}} \sim p_{1:n}}\left[\sup_{f \in \mathcal{F}} \frac{1}{n}\sum_{j=1}^{n}\frac{1}{m}\sum_{i=1}^{m}(\ell(f(\vec{x}'_{ij}), y'_{ij}) - \ell(f(\vec{x}_{ij}), y_{ij}))\right] \tag{14}$$

$$= \mathbb{E}_{S_{p_{1:n}} \sim p_{1:n}}\mathbb{E}_{S'_{p_{1:n}} \sim p_{1:n}}\mathbb{E}_{\vec{\sigma}}\left[\sup_{f \in \mathcal{F}} \frac{1}{n}\sum_{j=1}^{n}\frac{1}{m}\sum_{i=1}^{m}\sigma_{ij}(\ell(f(\vec{x}'_{ij}), y'_{ij}) - \ell(f(\vec{x}_{ij}), y_{ij}))\right] \tag{15}$$

$$\leq \mathbb{E}_{S'_{p_{1:n}} \sim p_{1:n}}\mathbb{E}_{\vec{\sigma}}\left[\sup_{f \in \mathcal{F}} \frac{1}{n}\sum_{j=1}^{n}\frac{1}{m}\sum_{i=1}^{m}\sigma_{ij}\ell(f(\vec{x}'_{ij}), y'_{ij})\right] \tag{16}$$

$$+ \mathbb{E}_{S_{p_{1:n}} \sim p_{1:n}}\mathbb{E}_{\vec{\sigma}}\left[\sup_{f \in \mathcal{F}} \frac{1}{n}\sum_{j=1}^{n}\frac{1}{m}\sum_{i=1}^{m}-\sigma_{ij}\ell(f(\vec{x}_{ij}), y_{ij})\right] \tag{17}$$

$$= 2\mathbb{E}_{S_{p_{1:n}} \sim p_{1:n}}\mathbb{E}_{\vec{\sigma}}\left[\sup_{f \in \mathcal{F}} \frac{1}{n}\sum_{j=1}^{n}\frac{1}{m}\sum_{i=1}^{m}\sigma_{ij}\ell(f(\vec{x}_{ij}), y_{ij})\right] \tag{18}$$

$$= 2\mathbb{E}_{S_{p_{1:n}} \sim p_{1:n}}[\mathcal{R}_{mn}(\mathcal{F})], \tag{19}$$

where the first inequality is from moving the supremum inside the expectation and the second is from subadditivity of suprema. Observing that the absolute difference of computing $\mathcal{R}_{mn}(\mathcal{F})$ on $\mathcal{S}$ and

$\mathcal{S}'$ cannot exceed $\frac{1}{mn}$, another application of McDiarmid's inequality tells us that with confidence at least $1 - \frac{\delta}{2}$

$$2\mathbb{E}_{S_{p_{1:n}} \sim p_{1:n}}[\mathcal{R}_{mn}(\mathcal{F})] \leq 2\mathcal{R}_{mn}(\mathcal{F}) + 2\sqrt{\frac{\ln(2/\delta)}{2mn}}. \tag{20}$$

Combining Equations 10 and 20 with the union bound concludes the proof. □

We now prove Theorem 1.

*Proof.* Theorem 2 tell us that with confidence at least $1 - \delta$,

$$\frac{1}{n}\sum_{j=1}^{n} L_{p_j}(f) \leq \hat{L}^{\mathcal{E}}(f) + 2\mathcal{R}_{mn}(\mathcal{F}) + 3\sqrt{\frac{\ln(2/\delta)}{2mn}}. \tag{21}$$

Thus, we must provide a (high confidence) upper bound on

$$L^{\mathcal{E}}(f) - \frac{1}{n}\sum_{j=1}^{n} L_{p_j}(f) \tag{22}$$

that holds uniformly for all $f \in \mathcal{F}$. We can use the same idea the proof for Theorem 2 to show how Rademacher complexity controls generalisation to novel domains, rather than novel instances within the same domain. Begin by letting $P = \{p_1, ..., p_n\}$ be an i.i.d. sample of $n$ domains from $\mathcal{E}$, and define

$$\Phi(P) = \sup_{f \in \mathcal{F}} L^{\mathcal{E}}(f) - \frac{1}{n}\sum_{j=1}^{n} L_{p_j}(f). \tag{23}$$

If we construct $P'$ by replacing any $p_j \in P$ with $p'_j \sim \mathcal{E}$, then we have $|\Phi(P) - \Phi(P')| \leq \frac{1}{n}$, so McDiarmid's inequality tells us that with confidence at least $1 - \frac{\delta}{2}$

$$\Phi(P) \leq \mathbb{E}_{p_{1:n} \sim \mathcal{E}}[\Phi(P)] + \sqrt{\frac{\ln(2/\delta)}{2n}}. \tag{24}$$

We proceed by bounding the expected value of $\Phi(P)$,

$$\mathbb{E}_{p_{1:n} \sim \mathcal{E}}\left[\sup_{f \in \mathcal{F}}\left(\mathbb{E}_{q \sim \mathcal{E}}[L_q(f)] - \frac{1}{n}\sum_{j=1}^{n} L_{p_j}(f)\right)\right] \tag{25}$$

$$= \mathbb{E}_{p_{1:n} \sim \mathcal{E}}\left[\sup_{f \in \mathcal{F}} \mathbb{E}_{q_{1:n} \sim \mathcal{E}}\left[\frac{1}{n}\sum_{j=1}^{n}\left(L_{q_j}(f) - L_{p_j}(f)\right)\right]\right] \tag{26}$$

$$\leq \mathbb{E}_{p_{1:n}, q_{1:n} \sim \mathcal{E}}\left[\sup_{f \in \mathcal{F}} \frac{1}{n}\sum_{j=1}^{n}\left(L_{q_j}(f) - L_{p_j}(f)\right)\right] \tag{27}$$

$$= \mathbb{E}_{p_{1:n}, q_{1:n} \sim \mathcal{E}}\mathbb{E}_{\vec{\sigma}}\left[\sup_{f \in \mathcal{F}} \frac{1}{n}\sum_{j=1}^{n}\sigma_j\left(L_{q_j}(f) - L_{p_j}(f)\right)\right] \tag{28}$$

$$\leq 2\mathbb{E}_{p_{1:n} \sim \mathcal{E}}\mathbb{E}_{\vec{\sigma}}\left[\sup_{f \in \mathcal{F}} \frac{1}{n}\sum_{j=1}^{n}\sigma_j L_{p_j}(f)\right] \tag{29}$$

$$= 2\mathbb{E}_{p_{1:n} \sim \mathcal{E}}\mathbb{E}_{\vec{\sigma}}\left[\sup_{f \in \mathcal{F}} \frac{1}{n}\sum_{j=1}^{n}\sigma_j \mathbb{E}_{(\vec{x},y) \sim p_j}[\ell(f(\vec{x}), y)]\right] \tag{30}$$

$$\leq 2\mathbb{E}_{p_{1:n} \sim \mathcal{E}}\mathbb{E}_{(\vec{x}_j, y_j) \sim p_j}\mathbb{E}_{\vec{\sigma}}\left[\sup_{h \in \mathcal{F}} \frac{1}{n}\sum_{j=1}^{n}\sigma_j \ell(f(\vec{x}_j), y_j)\right] \tag{31}$$

$$= 2\mathbb{E}_{p_{1:n} \sim \mathcal{E}}\mathbb{E}_{(\vec{x}_j, y_j) \sim p_j}[\mathcal{R}_n(\mathcal{F})], \tag{32}$$

where the first inequality comes from moving the supremum inside the expectation, $\vec{\sigma}$ is a vector of Rademacher random variables, the second inequality is due to the subadditivity of suprema, and the final inequality comes from moving the expectation outside of the supremum. Noting that replacing one of the $(\vec{x}_j, y_j)$ pairs in the final equality will result in $\mathcal{R}_n(\mathcal{F})$ changing by at most $\frac{1}{n}$, McDiarmid's inequality can be used to say with confidence $1 - \frac{\delta}{2}$ that

$$2\mathbb{E}_{p_{1:n} \sim \mathcal{E}} \mathbb{E}_{(\vec{x}_j, y_j) \sim p_j}[\mathcal{R}_n(\mathcal{F})] \leq 2\mathcal{R}_n(\mathcal{F}) + 2\sqrt{\frac{\ln(2/\delta)}{2n}}. \tag{33}$$

Combining Equations 21, 24, and 33 using the union bound completes the proof. $\qquad\square$

## B    PROOF OF COROLLARY 1

*Proof.* We begin with the expected excess risk, which can be bounded from above by

$$\mathbb{E}_S[L^{\mathcal{E}}(\hat{f}) - L^{\mathcal{E}}(f^*)] = \mathbb{E}_S[L^{\mathcal{E}}(\hat{f}) - \hat{L}^{\mathcal{E}}(\hat{f})] + \mathbb{E}_S[\hat{L}^{\mathcal{E}}(\hat{f}) - \hat{L}^{\mathcal{E}}(f^*)] + \mathbb{E}_S[\hat{L}^{\mathcal{E}}(f^*) - L^{\mathcal{E}}(f^*)]$$
$$\tag{34}$$

$$\leq \mathbb{E}_S[L^{\mathcal{E}}(\hat{f}) - \hat{L}^{\mathcal{E}}(\hat{f})] + \mathbb{E}_S[\hat{L}^{\mathcal{E}}(f^*) - L^{\mathcal{E}}(f^*)] \tag{35}$$

$$= \mathbb{E}_S[L^{\mathcal{E}}(\hat{f}) - \hat{L}^{\mathcal{E}}(\hat{f})] \tag{36}$$

$$= \mathbb{E}_S\left[L^{\mathcal{E}}(\hat{f}) - \frac{1}{n}\sum_{j=1}^n L_{p_j}(\hat{f})\right] + \mathbb{E}_S\left[\frac{1}{n}\sum_{j=1}^n L_{p_j}(\hat{f}) - \hat{L}^{\mathcal{E}}(\hat{f})\right], \tag{37}$$

where the inequality arises because, by definition, the *empirical* risk of $\hat{f}$ must be less than or equal to the empirical risk of the optimal model. The second equality comes from the fact that $f^*$ is determined independently of the particular training set that we sample, so $\mathbb{E}_S[\hat{L}^{\mathcal{E}}(f^*)] = L^{\mathcal{E}}(f^*)$. For the final equality: the first term can be bounded from above (with confidence $1 - \delta$) using the bound for the expected value of $\Phi(\mathcal{S})$ derived in the proof of Theorem 2; and the second term can be bounded from above (with confidence $1 - \delta$) using the bound for the expected value of $\Phi(P)$ derived in the proof for Theorem 1. Combining these two high confidence bounds using the union bound yields the result. $\qquad\square$

## C    DETAILED RESULTS

**Complexity Measurement for DomainBed Competitors**    In Figure 2 we illustrated the performance of DomainBed competitors in terms of their resulting model complexity by averaging over held-out domains for each dataset. In Figure 3, we present detailed results broken down by each held out dataset.

**Detailed results on each DG benchmark**    In Table 1 we reported results of our linear model competitors on DomainBed summarised over choice of held out domain. In Tables 2-7, we report the detailed results of accuracy and choice of $C$ paramater for each held out domain.

| Cross Validation | | 0 | 15 | 30 | 45 | 60 | 75 | Ave. |
|---|---|---|---|---|---|---|---|---|
| | | | | Raw pixel | | | | |
| Domain Wise | Acc | 0.577 | 0.785 | 0.799 | 0.783 | 0.769 | 0.586 | 0.716 (0.096) |
| | C | 256.0 | 128.0 | 128.0 | 128.0 | 64.0 | 256.0 | |
| Instance Wise | Acc | 0.582 | 0.774 | 0.788 | 0.783 | 0.765 | 0.579 | 0.712 (0.093) |
| | C | 1024.0 | 512.0 | 1024.0 | 1024.0 | 1024.0 | 1024.0 | |

Table 2: Accuracy and selected 'C' on Rotated-MNIST, using LinearSVC.

| Cross Validation | | A | C | P | S | Ave. |
|---|---|---|---|---|---|---|
| | | DINO ViT-B8 | | | | |
| Domain Wise | Acc | 0.877 | 0.676 | 0.971 | 0.446 | 0.743 (0.201) |
| | C | 0.25 | 0.125 | 0.25 | 0.125 | |
| Instance Wise | Acc | 0.866 | 0.620 | 0.891 | 0.445 | 0.706 (0.184) |
| | C | 4.0 | 8.0 | 8.0 | 2.0 | |
| | | DINO ViTS8 | | | | |
| Domain Wise | Acc | 0.864 | 0.666 | 0.948 | 0.441 | 0.730 (0.196) |
| | C | 0.5 | 0.0625 | 0.5 | 0.03125 | |
| Instance Wise | Acc | 0.867 | 0.676 | 0.858 | 0.464 | 0.716 (0.164) |
| | C | 2.0 | 4.0 | 8.0 | 1.0 | |

Table 3: Accuracy and selected 'C' on PACS, using LinearSVC.

| Cross Validation | | C | L | S | V | Ave. |
|---|---|---|---|---|---|---|
| | | DINO ViT-B8 | | | | |
| Domain Wise | Acc | 0.972 | 0.648 | 0.787 | 0.811 | 0.804 (0.115) |
| | C | 0.5 | 0.5 | 0.125 | 0.5 | |
| Instance Wise | Acc | 0.977 | 0.648 | 0.785 | 0.811 | 0.805 (0.117) |
| | C | 0.25 | 0.5 | 0.5 | 0.5 | |
| | | DINO ViTS8 | | | | |
| Domain Wise | Acc | 0.959 | 0.648 | 0.785 | 0.823 | 0.804 (0.111) |
| | C | 0.25 | 0.125 | 0.0625 | 0.25 | |
| Instance Wise | Acc | 0.959 | 0.651 | 0.789 | 0.823 | 0.806 (0.109) |
| | C | 0.25 | 0.5 | 0.25 | 0.25 | |

Table 4: Accuracy and selected 'C' on VLCS, using LinearSVC.

| Cross Validation | | A | C | P | R | Ave. |
|---|---|---|---|---|---|---|
| | | DINO ViT-B8 | | | | |
| Domain Wise | Acc | 0.748 | 0.512 | 0.832 | 0.850 | 0.736 (0.135) |
| | C | 1.0 | 1.0 | 1.0 | 0.5 | |
| Instance Wise | Acc | 0.730 | 0.514 | 0.833 | 0.846 | 0.731 (0.133) |
| | C | 4.0 | 2.0 | 2.0 | 4.0 | |
| | | DINO ViTS8 | | | | |
| Domain Wise | Acc | 0.740 | 0.496 | 0.820 | 0.834 | 0.722 (0.136) |
| | C | 1.0 | 0.25 | 0.5 | 0.5 | |
| Instance Wise | Acc | 0.726 | 0.508 | 0.824 | 0.833 | 0.723 (0.131) |
| | C | 2.0 | 1.0 | 1.0 | 2.0 | |

Table 5: Accuracy and selected 'C' on OfficeHome, using LinearSVC.

| Cross Validation | | aclass | escape | hilux | i3 | lexus | tesla | tiguan | tucson | x5 | zoe | Ave. |
|---|---|---|---|---|---|---|---|---|---|---|---|---|
| | | DINO ViT-B8 | | | | | | | | | | |
| Domain Wise | Acc | 0.979 | 0.980 | 0.967 | 0.997 | 0.959 | 0.981 | 0.942 | 0.986 | 0.975 | 0.954 | 0.972 (0.016) |
| | C | 1.0 | 1.0 | 1.0 | 1.0 | 2.0 | 1.0 | 2.0 | 1.0 | 1.0 | 2.0 | |
| Instance Wise | Acc | 0.922 | 0.991 | 0.910 | 0.998 | 0.902 | 0.960 | 0.937 | 0.983 | 0.976 | 0.955 | 0.953 (0.032) |
| | C | 128.0 | 64.0 | 128.0 | 64.0 | 256.0 | 128.0 | 256.0 | 128.0 | 128.0 | 128.0 | |
| | | DINO ViTS8 | | | | | | | | | | |
| Domain Wise | Acc | 0.961 | 0.959 | 0.959 | 0.971 | 0.958 | 0.964 | 0.933 | 0.979 | 0.957 | 0.814 | 0.945 (0.045) |
| | C | 0.125 | 0.5 | 0.25 | 0.5 | 0.5 | 0.5 | 0.25 | 0.25 | 0.25 | 0.25 | |
| Instance Wise | Acc | 0.914 | 0.969 | 0.952 | 0.996 | 0.966 | 0.838 | 0.843 | 0.989 | 0.810 | 0.851 | 0.913 (0.067) |
| | C | 1024.0 | 128.0 | 512.0 | 256.0 | 1024.0 | 64.0 | 512.0 | 128.0 | 512.0 | 128.0 | |

Table 6: Accuracy and selected 'C' on SVIRO, using LinearSVC.

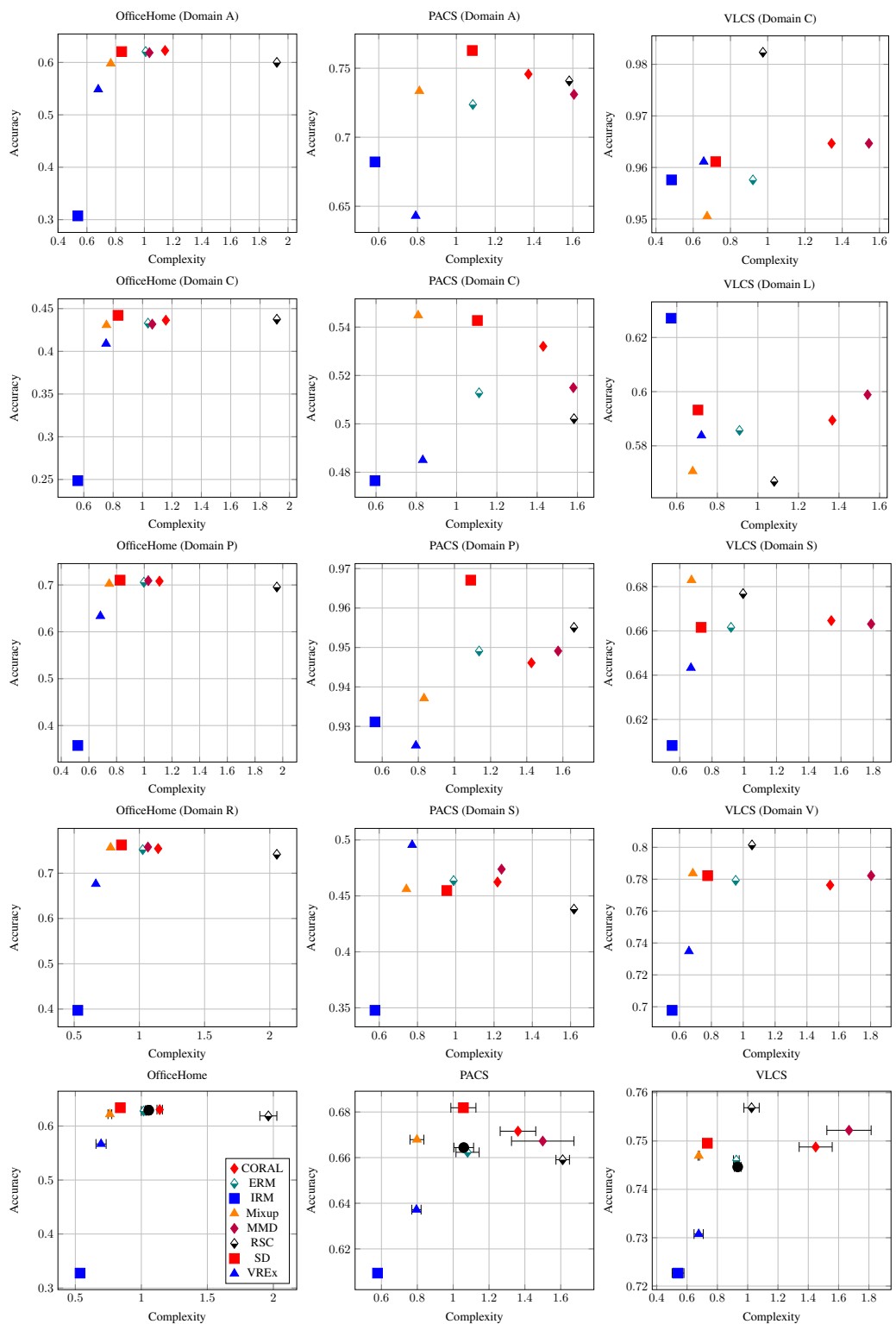

Figure 3: Performance of various neural network DG algorithms as a function of their measured model complexity after training using DomainBed. Breakdown by held-out dataset. The top four rows correspond to individual domains, the bottom row is the leave-one-domain-out cross-validation estimate of the DG performance, and horizontal error bars correspond to the standard deviation of the model complexities measured in each iteration of cross-validation.

| Cross Validation | | L100 | L38 | L43 | L46 | Ave. |
|---|---|---|---|---|---|---|
| DINO ViT-B8 | | | | | | |
| Domain Wise | Acc | 0.444 | 0.457 | 0.516 | 0.399 | 0.454 (0.041) |
| | C | 0.25 | 1.0 | 1.0 | 2.0 | |
| Instance Wise | Acc | 0.472 | 0.391 | 0.366 | 0.324 | 0.388 (0.054) |
| | C | 256.0 | 128.0 | 256.0 | 256.0 | |
| DINO ViTS8 | | | | | | |
| Domain Wise | Acc | 0.375 | 0.231 | 0.480 | 0.398 | 0.371 (0.090) |
| | C | 0.125 | 1.0 | 2.0 | 4.0 | |
| Instance Wise | Acc | 0.407 | 0.157 | 0.438 | 0.322 | 0.331 (0.109) |
| | C | 256.0 | 64.0 | 128.0 | 128.0 | |

Table 7: Accuracy and selected 'C' on Terra Incognita, using LinearSVC.

