# OpenReview forum: "Finding lost DG: Explaining domain generalization via model complexity"
_ICLR.cc/2022/Conference — ICLR 2022 Submitted_

### Official Review · Reviewer_YVsq · 2021-10-29

**Correctness:** 2
**Technical Novelty And Significance:** 2
**Empirical Novelty And Significance:** 1
**Recommendation:** 3
**Confidence:** 4

**Main Review:**


Strenths:
* The paper is one of the few early attempts to establish the theoretical framework of OOD/Domain Generalization.
* The connections with Rademacher complexity is interesting.

Flaws:
* The title seems to be a response to [3] but the evidence is not solid enough in comparison. I strongly recommend toning down and rename it to emphasize the theoretical contribution of this work.
* Some very relevant theoretical works are not cited and discussed like [1,2]. And indeed some findings are alreadly manifested in those works.
* The theoretical significance is not enough for publication. Domain generalization is unachivable without the formalization of distribution shifts but here only considers the risk over the mixtures of source domain, which is insufficient to provide out-of-distribution generalization guarantees.


[1] Towards a Theoretical Framework of Out-of-Distribution Generalization
[2] Empirical or Invariant Risk Minimization? A Sample Complexity Perspective
[3] In Search of Lost Domain Generalization


**Summary Of The Paper:**

This paper has several findings taking a bias-variance trade-off of DG performance, that explains existing DG algorithm performance variability. (ii) The complexity control strategy used to determine bias-variance trade-off is crucial in practice, with peak DG performance achieved when optimizing model complexity based on domain-wise validation. (iii) Regularisation required for optimal DG is greater than for conventional optimization for within-domain performance.

**Summary Of The Review:**

The paper has promising motivations but lacks of theoretical significance for DG community.

---

> ### Author Response · Authors · 2021-11-23
> **reply to reviewer4**
>
> Q1: Title oveclaim?
> A1: We are happy to change the title to simply “Explaining Domain Generalisation via Model Complexity” to reduce apparent overclaim.
>
> Q2: Relation to prior theory [1,2]?
> A2: [1] Does not have a sample complexity generalisation bound which we provide here. They analyse generalisation in terms of distribution distance, but this is not observable. In their experiments they approximate it with kernel density estimation, but the quality of their approximation is not well-justified. (Note also that our theory does not depend on the distance between the source and target distributions.) [2] Provides sample complexity bounds but they only apply to finite hypothesis classes. This means that it is not applicable to real-world model classes. In contrast, we are able to analyse models such as linear classifiers and neural networks with our Rademacher complexity based theory. We have added discussion of these papers to our related work section.
>
> Q3: Correctness of theory?
> A3: (i) The first statement “DG is unachievable without [assumption on] distribution shift” is wrong. Our proof shows it can be done in the general case, by assuming each domain is an IID sample from a prior distribution over domains, but without other assumptions about how the distributions for each domain are related. (ii) The second statement “[we consider] the risk over mixtures of source domain” is also wrong. We do not consider mixtures over source domains, we assume that domains seen at test time are novel domains sampled IID from the same distribution over domains that the training domains were sampled from. This is analogous to the classic machine learning setting where training data and testing data are both assumed to be IID samples from some data distribution.

---

> > ### Comment · Reviewer_YVsq · 2021-11-26
> > **Thanks for clarification**
> >
> > The formulation, especially the assumption that each domain is an IID sample from a prior distribution over domains, is quite douteful since it might renders a non-ood generalization problem when the test domain data is seen during training. It is because the test data is unseen during training that we need some assumptions on distribution shifts. Therefore, I don't buy the proposed theory with an assumption contradictory to the goal of DG.

---

> > > ### Author Response · Authors · 2021-11-27
> > > **IID vs sampling with replacement**
> > >
> > > It appears the reviewer is confusing two different concepts: (i) sampling with replacement from a finite set of domains; and (ii) IID sampling of domains from a distribution over a potentially infinite number of domains. If data were generated using (i), then it is true that training sets and test sets could overlap. However, we are assuming data is generated using (ii). As we have already stated, this is analogous to the near-universal assumption made by single-domain machine learning algorithms that data in training sets and test sets are IID samples from the same domain.
> > >
> > > This type of IID assumption in domain generalisation is not unusual or controversial. Seminal papers in the field have already used exactly this assumption when the DG problem was first conceived in its now standard form [BSL11], and also for some of the most influential theoretical work showing how DG can be (provably) accomplished for kernel machines [MBS13].
> > >
> > > [BLS11] Gilles Blanchard, Gyemin Lee, and Clayton Scott. Generalizing from several related classification tasks to a new unlabeled sample. NeurIPS, 2011.
> > >
> > > [MBS13] Krikamol Muandet, David Balduzzi, and Bernhard Schölkopf. Domain generalization via invariant feature representation. ICML, 2013.

---

### Official Review · Reviewer_jRak · 2021-10-29

**Correctness:** 3
**Technical Novelty And Significance:** 2
**Empirical Novelty And Significance:** 2
**Recommendation:** 5
**Confidence:** 4

**Main Review:**

### Strengths

**Problem setting.**  The setting is well-motivated, in that recent results have indicated that most DG algorithms cannot surpass the performance of ERM.  New theoretical insights are needed to understand why this happens, and indeed this work tries to fill this gap.

**Insights from Thm. 1**  The idea of treating the environments themselves as data is interesting, as it allows the derivation of this kind of bound which is agnostic of the form of $\mathcal{E}$.  I feel that the authors may consider taking this analysis further, as there may be new insights if e.g. $\mathcal{E}$ has some particular structure, or when the hypothesis class is linear (or has some structure).  In any case, Thm. 1 seems to be a nice starting point for this work.

**Interesting insights regarding accuracy vs. complexity.**  I quite liked the experiments in Section 3.1.  The takeaway seems to be that "the optimal regularization for [DG] is stronger than for [supervised learning]."  This indicates that the complexity of DG methods needs to be controlled more tightly than it does for regular supervised learning.  So at least in the linear setting, it seems to be the case that there is a clear trade-off between accuracy and complexity WRT test-time generalization.

### Weaknesses

**Confusing notation.**  There were several places in the paper where the notation/word-choice was confusing.  One point of confusion was that the risk $R_p$ and the Rademacher complexity $\mathcal{R}_m$ look rather similar. Perhaps the risk could be changed to $L_p$, which is the notation used in [Shalev-Shwartz & Ben-David, 2014].  The word-choice used to differentiate supervised learning from domain generalization was confusing.  If my understanding is correct, the former is referred to as supervised learning, seen-domain, and within-domain whereas the latter is referred to as novel-domain, cross-domain, and domain generalization.  Perhaps choosing just one term to refer to each of these would make the paper easier to read.  A further point

**Typos.**  There are a number of typos.  Both "regularizers" (page 1) and "regularisation" (page 2) are used; the paper should be consistent in choosing the American or British spelling.  The dataset is Terra-Incognito (page 5).

**The formulation of the DG problem.**  The authors assert that the goal of DG is to solve the following optimization problem:

$$ \min_{f\in\mathcal{F}} \mathbb{E}_{p\sim\mathcal{E}} [R_p(f)] $$

where $\mathcal{E}$ is a distribution over distributions $p$.  This is not the only formulation for domain generalization.  Indeed, some would argue that this is not the most standard formulation for the problem.  Another formulation was considered in [Arjovsky et al., 2019], wherein the authors advocate for the following "worst-case" problem:

$$ \min_{f\in\mathcal{F}} \max_{p\in\Delta\} R_p(f)$$

where $\Delta$ is something like the support of $\mathcal{E}$ in the notation of this paper.  The reason I feel that this is important is that the authors make assertions like this:

> "any future DG method that claims to be better than ERM should..."

These assertions are only relevant to the first "average-case" definition of the DG problem, and do not necessarily apply to other common formulations, e.g. the "worst-case" formulation shown above.  One way to remedy this would be to move the proofs to the appendix and to add a more detailed discussion of related work and of past formulations and approaches for DG.

**Lack of sufficient detail.**  There were numerous places where I thought the paper could have benefitted from providing more detail.  Here are some questions I had while reading, all of which could be addressed by adding details:
* What do the authors mean by the "domain-shift phenomena?"  Of late there has been quite a bit of work on domain shift in the sense of Section 1.8 in [Quiñonero-Candela, et al., 2009], which defines domain shift to be a particular form of covariate shift.  However, it seems that the authors may be using domain-shift to informally refer to the problem of domain generalization, wherein there can be arbitrary shifts in the joint distribution over (X,Y) from domain to domain.  Resolving this confusion would make the problem setting clearer.
* Define what is meant by "data in the wild" (page 1).
* A recurring question I had was the following: What do the authors mean when they say that X is a **causal factor** in the success or failure of DG methods.  Is "causal" meant informally?  Or does it mean something more technical vis-a-vis the field of causal inference.  This term is used repeatedly, and after reading the paper several times, I still am not sure what it means.
* The authors talk quite a bit about the "performance variability" (page 2) or "erratic performance" (page 1) "unreliable" performance (page 6) of existing DG methods.  It's unclear whether we should read this as "poor performance" or "non-stable performance."  I find this distinction important because the results of [Gulrajani & Lopez-Paz, 2020] indicate that on the contrary, the collective performance of DG baselines is actually quite stable in the sense that they almost all algorithms do more or less the same as ERM.  Now to be fair, this performance isn't all that great, given that there is a very large gap between in-distribution and out-of-distribution performance of each of these methods.  Thus, I think that there needs to be a bit more subtlety in how the authors specify the weakness of the DG algorithms that they want to improve.  Additionally, I think that this perspective that ERM is optimal for this "average-case" is not new; [Rosenfeld et al., 2021] also advocates for this view, so it may be good to cite their work here.
* At the start of the experiments, the authors say they will work with linear SVM.  But then, the authors say that they train linear SVC (which I assume is the classification analog of SVM, although it's never stated).  This should be clarified.
* What do the authors mean by a "dynamic range of accuracy" (page 7)?  What is "dynamic" about it?

**Unclear explanation of experiments.** I didn't understand the result sin Section 3.3.  What is the difference between instance-wise and domain-wise validation?  This is never explained from what I can tell.  Does this simply mean in-domain validation vs. cross-domain validation?  It's also unclear to me what claim in [Gulrajani & Lopez-Paz, 2020] the authors of this paper are disputing at the bottom of page 7.  From my understanding, [Gulrajani & Lopez-Paz, 2020] argues that DG algorithms should be responsible for specifying a hyperparameter selection criteria.  How do these results refute this claim?

**Claims in the abstract.**  Some of the claims in the abstract are somewhat dismissive of past work.  They say that "most of the [past work in DG] is empirical, as the DG problem is hard to model formally."  They go on to say that they do something more principled with a learning-theoretic analysis of the problem.  This dismisses the large and growing body of work that has sought to understand the theoretical underpinnings of the DG problem.  Indeed, many of the proposed algorithms for DG are past on concrete theory; the fact that they don't in general significantly improve over ERM is a separate issue.  Furthermore, in the end, the authors actually do something quite empirical, in the sense that they extract features using a deep classifier and look at the performance of small neural networks on DomainBed.  This to me seems no less "empirical" than many of the other works that have sought to study this problem.  So from my reading of this paper, the authors also fall into the pitfall that they claim to identify early on in the paper.  More than anything else, I think this is an issue with how the idea proposed in this paper is discussed, rather than being a critique of the proposed method itself.  So I would recommend softening these claims and expanding the related works section, given that there is a great deal of related work in DG that has been left out.

**Formal-ness of the proofs.**  In my opinion, the proofs here are presented too informally.  The authors use terms like "ghost sample" without defining or describing them for the reader.  I think that eq(7) should be proved formally, rather than hand-waving over the argument (whether or not the authors feel it is trivial).  The same can be said for Thm. 2 (which I would call Corollary 2 rather than Theorem 2), for which the proof is more of a sketch.

**Softening some of the claims.**  The authors have a section titled called "Solving Domain-Bed with Linear Models."  I think that this is misleading, as the authors are not "solving" DomainBed.  "Solving" would indicate that they found a way to achieve perfect domain generalization by generalizing OOD.

### References

[Shalev-Shwartz & Ben-David, 2014] Shai Shalev-Shwartz and Shai Ben-David. Understanding machine learning: From theory to algorithms. Cambridge university press, 2014.

[Arjovsky et al., 2019] Arjovsky et al. "Invariant risk minimization." arXiv preprint arXiv:1907.02893 (2019).

[Quiñonero-Candela, et al., 2009] Joaquin Quiñonero-Candela, et al., eds. Dataset shift in machine learning. Mit Press, 2009.

[Gulrajani & Lopez-Paz, 2020] Ishaan Gulrajani and David Lopez-Paz. "In search of lost domain generalization." arXiv preprint arXiv:2007.01434 (2020).

[Rosenfeld et al., 2021] Elan Rosenfeld, Pradeep Ravikumar, and Andrej Risteski. "An online learning approach to interpolation and extrapolation in domain generalization." arXiv preprint arXiv:2102.13128 (2021).

**Summary Of The Paper:**

This paper considers the problem of domain generalization (DG), wherein predictors are trained on a related set of training domains and evaluated on an unseen test domain.  The authors first present a learning-theoretic bound on the performance of an average-case formulation for DG, and then present a set of experiments that consider the trade-off between complexity and out-of-distribution (OOD) performance.  The experiments indicate that such trade-offs exist for linear models, and there is also evidence that the trade-off persists for shallow neural networks.

**Summary Of The Review:**

On the one hand, I think that this paper offers an interesting perspective for thinking about DG.  The theoretical work gives a new way of thinking about the accuracy-complexity trade-off in DG, and this leads to some interesting experiments.  However, I think that the paper tends to overclaim, in particular with regard to "solving DomainBed" and the implications of its theory for ERM.  Furthermore, I think that the writing is particularly unclear, and that the claims made in the introduction are not fully validated (e.g. the claims about the "causal" factor behind poor performance in DG).  Overall, despite the interesting insights, I think that this paper is not yet ready, and so I recommend that it not be accepted.

---

> ### Author Response · Authors · 2021-11-23
> **reply to reviewer3**
>
> Q1: Confusing notation?
> A1: We have updated our notation to reduce some of this confusion.
>
> Q2: Is the manuscript written in British English or American English?
> A2: We have made the spelling consistent.
>
> Q3: Why expected error and not worst-case error?
> A3: Most general purpose DG methods are concerned with the expected error, not the worst-case error. Prior work based on methods from causal inference typically addresses the worst-case problem setting (e.g., [ABGL19]). However, many of the general purpose methods either directly explicitly address the average-case problem [BLS11, MBS13] or at least put emphasis on comparing performance in the average case when evaluating proposed methods [GL21]. We have expanded our related work section to provide additional context for our work, and alternative formulations that are beyond the scope of our paper.
>
> Q4: What is meant by domain shift?
> A4: Most DG methods assume p(y|x) remains constant and only p(x) changes, as this is the type of distribution shift encountered in popular DG benchmarks. However, it’s worth noting that our theoretical analysis applies to both situations: change in p(x) and change in p(y|x). We have added some discussion about this in the paper.
>
> Q5: What is meant by causal factor?
> A5: Figure 1, we controlled model complexity and demonstrated that it determines model accuracy in the DG setting, which is what we meant by causal face. Figure 2 is observational rather than interventional, but the result is consistent with Figure 1. We have toned down the writing to avoid overclaiming.
>
> Q6: What is meant by performance variability?
> A6: We are referring to the observation of [GL21], where the rankings of DG methods across different benchmarks can be said to be erratic or unreliable because they typically all have very similar accuracy. They observed that no model in their suite performed consistently better than ERM across all benchmarks.
>
> Q7: How is this different to Rosenfeld et al. (2021)?
> A7: The Rosenfeld paper deals with online DG (where each time-step corresponds to getting a new domain), and the domain at time t is chosen by an adversary. They analyse this game for a finite number of timesteps, but they assume each domain has an infinite amount of data. They also put some limitations on the adversary. E.g., must choose a domain that is a convex combination of a finite number of pre-determined domains. In contrast, our theoretical analysis is in the more realistic setting where one has a finite amount of data per domain, and the domains we consider are not limited to convex combinations of a set of prototypical domains.
>
> Q8: Instance-wise vs domain-wise validation?
> A8: Yes, this corresponds to using in-domain validation data vs. cross-domain data. We have revised the paper to explain this more clearly.
>
> Q9: What result of [GP21] do we dispute?
> A9: [GP21] observed that instance-wise was a better validation criterion than domain-wise. We have shown the opposite is the case when using better hyperparameter tuning.
>
> Q10: Abstract Claims?
> A10: Thanks. Our focus is on general purpose DG methods, rather than the causal inference family of methods that each make specific assumptions about the underlying family of structural equation models (see revised discussion in related work), and we believe the former has much less theoretical work. But we agree the previous phrasing was overly strong, and we have reduced the scope of our claim.
>
> Q11: Relationship to prior DG theory work?
> A11: We have expanded the RW section to better contextualize our work wrt. existing theory and explain the particular type of DG theory that we are going for to clarify the breadth of our claims (ie: Focus on the general purpose rather than causal inference setting). We are happy to cite any additional theory work that we are missing if you have any pointers.
>
> Q12: Formalness of proofs.
> A12: Thanks. We’ve revised them significantly.
>
> Q13: Linear models on DomainBed.
> A13: Thanks for the comment. We meant it in the sense of “applying to” domain bed, including hyperparameter selection. Not in the sense of  “problem solved”. We are sorry for the confusion and have changed the phrasing to avoid the appearance of overclaim.
>
> [ABGL19] Martin Arjovsky, Léon Bottou, Ishaan Gulrajani, David Lopez-Paz. Invariant Risk Minimization. 2019.
> [BLS11] Gilles Blanchard, Gyemin Lee, and Clayton Scott. Generalizing from several related classification tasks to a new unlabeled sample. NeurIPS, 2011.
> [MBS13] Krikamol Muandet, David Balduzzi, and Bernhard Schölkopf. Domain generalization via invariant feature representation. ICML, 2013.
> [GL21] Ishaan Gulrajani, David Lopez-Paz. In Search of Lost Domain Generalization. ICLR, 2021.

---

> > ### Comment · Reviewer_jRak · 2021-12-06
> > **Thanks for your reply + further comments on my end**
> >
> > Thanks for your detailed response!  The proofs seem to be much more clearly presented, and I think that the paper reads better in this current version.  Removing the claims about finding the so-called "causal factor" have definitely improved the clarify of the claims.  The expansion of the related work section greatly improves the paper in my opinion as well.
> >
> > I do agree with the other reviewers that the theory does not apply to the majority of datasets that are run, in the sense that one would need a large number of training domains to benefit from the guarantees provided by the theorems.  I think that the authors should discuss this more in the main text.  I am also not convinced that "most general purpose DG methods are concerned with the expected error."  A number of notable works which take the worst-case perspective are as follows:
> >
> > https://arxiv.org/pdf/2106.06607.pdf
> > https://proceedings.neurips.cc/paper/2021/file/a8f12d9486cbcc2fe0cfc5352011ad35-Paper.pdf
> > https://proceedings.neurips.cc/paper/2018/file/1d94108e907bb8311d8802b48fd54b4a-Paper.pdf
> > https://arxiv.org/pdf/1907.02893.pdf
> > https://arxiv.org/pdf/2106.09913.pdf
> >
> > Extending the theory to handle this worst-case problem formulation would constitute a more significant contribution, as I still firmly believe that the average-case analysis is only one of the possible interpretations of the DG problem setting.
> >
> > All this being said, I think that the paper has been improved and the authors addressed many of my concerns.  In response I will raise my score.

---

### Official Review · Reviewer_ZKqa · 2021-10-30

**Correctness:** 3
**Technical Novelty And Significance:** 2
**Empirical Novelty And Significance:** 3
**Recommendation:** 5
**Confidence:** 4

**Main Review:**

# Strong point

- Modeling domain generalization performance from a domain-complexity view is novel. The paper shows an interesting trade-off between model complexity and training loss in a derived upper bound.

- Extensive experiments showcase the validity of the theory. A better model selection strategy is proposed.

- It is nice to see that the optimal regularization of across-domain acc is consistently larger than the one of within-domain acc in Fig (1).

# Weakness

- The demonstration of the trade-off in Theorem 1 is nice. The results of the arguments rely on concentration bounds such as McDiarmid's inequality, which is typically used for large $n$. However, in reality, the number of training domains is small, e.g., 4~5 training domains. The bound seems vacuous when $n$ is small.

- Another caveat is that theorem 1 bound the average risk w.r.t some environmental prior $\mathcal{E}$. One would imagine that the average risk is dominated by the risk of environments in the high probability regions of the prior. The model can exploit the environmental-specific feature in those environments regardless of loss of the environments in the low probability region, which is the motivation of many existing DG works.

- According to Eq (16), it seems that the Rademacher term $\mathcal{R}_n(F)$ is the Rademacher complexity but not empirical Rademacher complexity. In addition, the notations are confusing. The underlying joint distributions for (x,y) are different for the two Rademacher complexity terms in Eq (7).

- It seems that there is no clear trait of the trade-off in Fig (2). One interesting observation is that IRM has the smallest model complexity and held-out domain accuracy in the baselines, which contradicts existing works. [1] theoretically shows that it is easy to fit the IRMv1 objective while behaving like ERM on held-out domains. The empirical results in [2], sec 3.2 also suggests that the IRMv1-trained model behaves like the ERM-trained model. Is such discrepancy caused by disparate architectures (MLP versus ResNet)? Also, the held-out accuracy in VLCS datasets of ERM models seems to decrease with model complexity.


# Minors

- Across-domain accuracy and held-out domain accuracy are used interchangeably.
- What is ``instance-wise" model selection? The average loss of data instances in training domains?

[1]: Elan Rosenfeld, P. Ravikumar, and Andrej Risteski.  The risks of invariant risk minimization. ArXiv, abs/2010.05761, 2020

[2]:Yilun Xu and T. Jaakkola. Learning Representations that Support Robust Transfer of Predictors. ArXiv, abs/2110.09940, 2021

**Summary Of The Paper:**

This paper presents a theory of domain generalization based on statistical learning theory (Rademacher complexity) and demonstrates a trade-off between training loss and model complexity. They show that existing methods are actually controlling the model complexity. Based on the analysis, the authors argue that proper model selection is critical for complexity control. Instead of hyper-parameter search, they propose to use domain-wise cross-validation as the model selection strategy. Experiments on the DomainBed benchmark show the effectiveness of the proposed method.

**Summary Of The Review:**



To summarize above, I find the theoretical results are somewhat unsatisfying due to limited domain numbers in practice. The experimental parts report results on small models, which contradicts some existing results on larger ones. All in all, I find the paper in its current form to be somewhat below the standard for ICLR acceptance, unless the authors address some of the points above.

---

> ### Author Response · Authors · 2021-11-23
> **reply to reviewer2**
>
> Q1: Are the bounds vacuous for real DG problems?
> A1: See R1Q2.
>
> Q2: Environmental prior.
> A2: Sorry but we do not understand this comment. If you could elaborate your point here and provide references to the existing DG works you refer to, we would be very happy to discuss this point.
> Please note that our bound needs no assumption on the shape of the prior.
>
> Q3: Rademacher complexity or empirical Rademacher complexity?
> A3: We use empirical Rademacher complexity throughout. We have changed the notation and proofs to clarify this.
>
> Q4: Tradeoff in Fig 2? IRM Results in Fig 2
> A4: We now illustrate the tradeoff between overfitting and underfitting regimes using polynomial regression curves. In the underfitting regime (left) performance is sharply lower, in the overfitting regime (right) performance decays slowly, with best performance somewhere in the middle.
> With regards to IRM: Short answer: The apparent discrepancy is not due to MLP vs ResNet and we do not contradict the prior work. We now include a version of IRM that is tuned to perform closer to ERM by reducing its regularisation strength. Long answer:It’s important to note that these models are run without careful hyperparameter tuning in Fig 2. Depending on hyperparameters, models will achieve different complexities (e.g., as we showed in Fig 1). So the specific placement of any given model (such as IRM, ERM, etc) on the graph will depend on hyperparameter choice. Our point is not about the absolute position of any given model, but the fact that when complexity is allowed to vary by any means (due to regulariser strength, training iterations, choice of ERM vs IRM vs REX etc), the resulting complexity  (x-axis) determines the performance (y-axis).

---

> > ### Comment · Reviewer_ZKqa · 2021-12-01
> > **Thanks for the response**
> >
> > I thank the authors for their response. The notations and definitions are now more consistent.
> >
> > My second question is related to the environmental prior $\mathcal{E}$ over the domains. Theory appears to be unable to guarantee the accuracy of tail domains, which have low probability density in $\mathcal{E}$. It would become worse if few environments make up a large proportion of the prior $\mathcal{E}$. However, the current framework does not assume any structure in the prior.
> >
> > I somewhat doubt the answer to Q4. The IRMv1 baseline is insensitive to its hyper-parameter in the over-parameterized regime. Indeed, it's observed that an ERM-trained model can have extremely small gradient penalty in IRMv1.

---

### Official Review · Reviewer_LqhM · 2021-11-05

**Correctness:** 3
**Technical Novelty And Significance:** 4
**Empirical Novelty And Significance:** 4
**Recommendation:** 8
**Confidence:** 4

**Main Review:**

**Strengths**
1. Generalization bound for DG setting
2. Empirical experiments support that ood generalization requires smaller model complexity
3. Propose cross-domain validation as a metric to optimize DG models

**Weaknesses**
1. No discussion on the size of n in theory section. Usually n is very small.
2. Overclaiming model complexity as the main reason explaining the DG results.

I feel that it is a good paper shedding light on ill-formed DG problem. I summarize my feedback for the authors below:

The generalization bound is a nice characterization to show that we may need additional regularization for OOD generalization. It may be good to discuss this relative to other papers that have made similar claims about regularization (e.g., causal regularization by Janzing https://arxiv.org/abs/1906.12179). The conclusion on regularization does not seem a novel idea to me, but I do like the theoretical rigor to support it.

The discussion after Theorem 2, however, misses the point of what empirical domain generalization is about. With enough diverse domains (e.g., 100-1000), of course any method (including ERM) would be good. But the tasks typically have 4-5 domains which makes generalization harder. Given extreme domain shift, DG methods can easily outperform ERM (see e.g., Arjovsky et al. IRM paper, or Mahajan et al. MatchDG paper). The only issue is that in the DG benchmark datasets (and many real-world settings with small domain shifts), ERM seems to perform better. So while Theorem 2 may be true, I do not think it has much of an insight for empirical DG. Perhaps the point that authors want to make is that ERM would work well for all settings, while a DG algorithm would work well in some datasets, but not all (based on its assumptions).

I appreciate the empirical results. Nice formulation to use linear models over pre-trained representation, that helps to confirm the model complexity hypothesis. For Figure 2, however,  I would like to caution that the results only show that model complexity is _one_ of the factors that contribute to  ood accuracy. It is no surprise that ood accuracy increases up to a certain model complexity and then decreases. I would suggest the authors to stick to the main claim (ood gen requires smaller model complexity than iid gen). The claim that complexity determines the OOD generalization is not supported by the experiments. Are the authors saying that a more complex model can _never_ achieve higher OOD generalization? That would be a very strong statement.

I would also tone down section 3.3: "solving domainbed". All the numbers reported are lower than state-of-the-art. The paper just compares a linear model with different validation schemes.

In theorem 1, did not see a derivation for the last term O(sqrt(ln(1/\delta)/n) in the proof. How is that derived?


**Summary Of The Paper:**

In domain generalization, recent work has shown that ERM has comparable out-of-domain accuracy to state-of-the-art DG methods. The paper aims to explain this result through model complexity. The main argument is that ood generalization requires a smaller model complexity. The upshot is that DG methods can utilize cross-domain validation to obtain better generalizing models.

**Summary Of The Review:**

Good paper explaining the recent empirical results against DG algorithms

---

> ### Author Response · Authors · 2021-11-23
> **reply to reviewer1**
>
> Q1: How does this work relate to Janzing (2019)?
> A1: There are several significant differences to the scope of our work compared to Janzing. Janzing assumes access to infinite data in each domain (we assume a finite training set), uses a non-standard measure of model complexity (we use Rademacher complexity), and restricts analysis to problems with a specific causal structure, while we make no causal structure assumption. The former means that the wide range of existing tools to measure or constrain Rademacher complexity for diverse model classes can be used with ours, but not Janzing’s. The latter means that our analysis is more generally applicable to diverse problems. Moreover, it is unclear whether the object they bound (expected risk under the interventional distribution) is equivalent to what we bound (expected risk across all domains). We have added discussion of this paper (and others) to our related work section.
>
> Q2: When is Theorem 2 useful, since DG problems often have a small number of domains?
> A2: This is primarily a weakness in some DG benchmarks such as PACS and VLCS, not a weakness of our theoretical contributions. Newer benchmarks such as WILDS contain datasets with orders of magnitudes more domains. It is true that some DG methods derived using assumptions about the underlying structural equation model or causal diagram associated with a particular DG problem instance can generalise to new domains with only a few training domains. However, many DG methods are not rooted in this type of analysis, and are intended for general purpose DG where one cannot make any strong assumptions about the underlying causal structures. It has been shown experimentally that these more general methods typically have no benefit over ERM [GL21], and it has also been shown theoretically that methods such as IRM do not provide an advantage over ERM in the general case [RRR21]. We have updated the discussion after Theorem 2 to make it clearer how our analysis relates to methods based on causal inference.
>
> Q3: Are the authors saying that a more complex model can never achieve higher OOD generalization?
> A3: No. We claim that model complexity is ‘a’ key factor in DG performance, not ‘the only factor’, or ‘the’ key factor. Note that our bound in Theorems 1 and 2 relate model complexity to the worst-case generalisation—i.e., they are true even the DG problem with the most pathological distribution shift. One can think of appropriate control of model complexity as a sufficient condition for DG, not a necessary condition.
>
> Q4: Sec 3.3 title.
> A4: Thanks for the comment. We meant it in the sense of “applying to” domain bed, including hyperparameter selection. Not in the sense of  “problem solved”. We have changed the phrasing to avoid the appearance of overclaim.
>
> Q5: Where do the O(...) terms in Theorem 1 come from?
> A5: These arise due to McDiarmid’s inequality. We have added more detail to the proof to clarify this.
>
> [GL21] Ishaan Gulrajani, David Lopez-Paz. In Search of Lost Domain Generalization. ICLR, 2021.
> [RRR21] Elan Rosenfeld, Pradeep Kumar Ravikumar, Andrej Risteski. The Risks of Invariant Risk Minimization. ICLR, 2021.

---

> > ### Comment · Reviewer_LqhM · 2021-11-30
> > **thanks for these clarifications**
> >
> > I appreciate the answer to 2.

---

### Author Response · Authors · 2021-11-29
**General Response to Reviewers**

We thank the reviewers for the time they have put into reviewing the paper and their helpful feedback. In general, the reviewers all find our novel link between DG performance and Rademacher complexity interesting, as well as the practical consequences for hyperparameter tuning.

The main weaknesses identified were related to presentation and a lack of contextualisation of our contributions within the wider DG literature. We have addressed these aspects quite considerably in our revised manuscript by expanding the related work section, adding more detail to the proofs, and clarifying some of the experimental setup issues.

In light of this, we would appreciate if the reviewers would reconsider their final scores.

---

### Decision · Program_Chairs · 2022-01-20

**Decision:**

Reject

**Comment:**

This paper provides a learning theoretic account of domain generalization in which domains themselves are treated as data, generated from some domain generating distribution. All of the reviewers were positive about this approach and found it interesting. There were, however, a couple of critiques raised by reviewers that lead me to recommend that it is rejected:

- the theory provided in this paper does not remotely apply to the datasets that are used in the experiments. While, I agree with one of the author responses that DG benchmarks exist with many domains, DomainBed has very few domains, and it's not clear that their theory is a remotely satisfactory account of the experimental results presented in the paper.
- Despite some back and forth on the wording and positioning of the paper, I think the writing still does not give enough credit to worst-case analyses of DG.